# HR-pQCT for the Evaluation of Muscle Quality and Intramuscular Fat Infiltration in Ageing Skeletal Muscle

**DOI:** 10.3390/jpm12061016

**Published:** 2022-06-20

**Authors:** Simon Kwoon-Ho Chow, Marloes van Mourik, Vivian Wing-Yin Hung, Ning Zhang, Michelle Meng-Chen Li, Ronald Man-Yeung Wong, Kwok-Sui Leung, Wing-Hoi Cheung

**Affiliations:** 1Musculoskeletal Research Laboratory, Bone Quality and Health Centre, Department of Orthopaedics and Traumatology, The Chinese University of Hong Kong, Hong Kong; vivi@ort.cuhk.edu.hk (V.W.-Y.H.); ningzzz@stanford.edu (N.Z.); mclimichelle@link.cuhk.edu.hk (M.M.-C.L.); ronald.wong@cuhk.edu.hk (R.M.-Y.W.); ksleung@cuhk.edu.hk (K.-S.L.); louis@ort.cuhk.edu.hk (W.-H.C.); 2Orthopaedic Biomechanics, Department of Biomedical Engineering, Eindhoven University of Technology, 5600 MB Eindhoven, The Netherlands; m.v.mourik@tue.nl

**Keywords:** intramuscular infiltration, sarcopenia, HR-pQCT, aged skeletal muscle, animal model

## Abstract

Myosteatosis is the infiltration of fat in skeletal muscle during the onset of sarcopenia. The quantification of intramuscular adipose tissue (IMAT) can be a feasible imaging modality for the clinical assessment of myosteatosis, important for the early identification of sarcopenia patients and timely intervention decisions. There is currently no standardized method or consensus for such an application. The aim of this study was to develop a method for the detection and analysis of IMAT in clinical HR-pQCT images of the distal tibia to evaluate skeletal muscle during the ageing process, validated with animal and clinical experimentation. A pre-clinical model of ovariectomized (OVX) rats with known intramuscular fat infiltration was used, where gastrocnemii were scanned by micro-computed tomography (micro-CT) at an 8.4 μm isotropic voxel size, and the images were analyzed using our modified IMAT analysis protocol. IMAT, muscle density (MD), and muscle volume (MV) were compared with SHAM controls validated with Oil-red-O (ORO) staining. Furthermore, the segmentation and IMAT evaluation method was applied to 30 human subjects at ages from 18 to 81 (mean = 47.3 ± 19.2). Muscle-related parameters were analyzed with functional outcomes. In the animal model, the micro-CT adipose tissue-related parameter of IMAT% segmented at −600 HU to 100 HU was shown to strongly associate with the ORO-positively stained area (r = 0.898, *p* = 0.002). For the human subjects, at an adjusted threshold of −600 to −20 HU, moderate positive correlations were found between MV and MD (r = 0.642, *p* < 0.001), and between MV and IMAT volume (r = 0.618, *p* < 0.01). Moderate negative correlations were detected between MD and IMAT% (r = −0.640, *p* < 0.001). Strong and moderate associations were found between age and MD (r = −0.763, *p* < 0.01), and age and IMAT (r = 0.559, *p* < 0.01). There was also a strong correlation between IMAT% and chair rise time (r = 0.671, *p* < 0.01). The proposed HR-pQCT evaluation protocol for intramuscular adipose-tissue produced MD and IMAT results that were associated with age and physical performance measures, and were of good predictive value for the progression of myosteatosis or sarcopenia. The protocol was also validated on animal skeletal muscle samples that showed a good representation of histological lipid content with positive correlations, further supporting the clinical application for the rapid evaluation of muscle quality and objective quantification of skeletal muscle at the peripheral for sarcopenia assessment.

## 1. Introduction

Sarcopenia is a disease characterized by the progressive loss of skeletal muscle mass and strength [1], which is highly associated with unintentional fall events and fragility fractures in older people [2]. The early detection of sarcopenia would enable personalized treatment decisions for the prevention of fragility fracture. Myosteatosis is the infiltration of fat in skeletal muscle during sarcopenia onset [1,3] with the overall volume of muscle remaining unchanged and the decrease in lean muscle, resulting in an unnoticeable loss in muscle strength and eventually dynapenia (i.e., loss of muscle strength). Delmonico et al. reported that 35.5% to 74.6% of intermuscular fat increase was detected in men aged 70 to 79 in 5 years’ time, while a 16.8% to 50.0% increase was detected in women. The fat tissue content within skeletal muscle was also found to be associated negatively with muscle strength in elderly people [4]. These observations are also well supported by a sarcopenic animal model showing that increased intramuscular fat infiltration has direct consequences on muscle strength and physical performance [5]. The exact mechanism that leads to myosteatosis is not fully understood; however, it is believed that disuse, sex steroid depletion, and altered leptin signaling are associated with lipid accumulation in elderly people. Furthermore, it is considered as one of the pathological factors of sarcopenia [6,7,8], affecting approximately 10% of both males and females in the world [9], leading to poor balancing abilities and a substantial increase in fall risks and rates of fragility fracture (to as much as 1.87 times higher) [10,11].

The imaging and quantification of intramuscular adipose tissue (IMAT) are therefore important to study and understand this factor during skeletal muscle ageing. Various advancements in imaging modalities have made this task possible [12]. Magnetic resonance imaging (MRI) [13,14] and computed tomography (CT) [15,16] are effective gold standards to accurately discriminate adipose tissue from skeletal muscles. However, the clinical application of these imaging techniques are technically limited by the lack of a standardized evaluation protocol [17] and practically limited by its availability and long waiting time at various hospital settings [18]. Alternatively, dual energy absorptiometry (DXA) and bioimpedance analysis (BIA) are suggested as alternatives to quantitatively estimate whole body lean and fat muscle contents [6,19]. However, the resolution of these techniques is limited by their insufficient resolving power to discriminate IMAT.

High-resolution peripheral quantitative computed tomography (HR-pQCT) is an advanced method of evaluating microstructures of the bone [20,21,22] or muscle at resolutions of up to 41 microns with an effective radiation dose of less than 5 μSv per scan as compared to 50–150 μSv per chest X-ray. HR-pQCT is able to evaluate soft tissues and discriminate adipose tissues from lean skeletal muscles to evaluate a number of parameters important for the estimation of adipose tissues in the region of interest. These parameters include muscle density (MD) and inter- or intramuscular adipose tissue (IMAT), important for the assessment of muscle quality. Furthermore, the HR-pQCT also has the advantage of generating three-dimensional volumetric reconstructions in a relatively short scanning time compared to DXA and BIA. Therefore, it is suggested that HR-pQCT is a feasible method for the evaluation of muscle quality and intramuscular fat infiltration during the progression of sarcopenia. Here, we describe the application of a method for the detection and analysis of intramuscular adipose tissue (IMAT) in clinical HR-pQCT images, to supplement a previously reported method that quantified “inter”-muscular adipose tissue [23]. This method was further pre-clinically validated with animal experimentation by in vivo micro-CT and histological analysis.

## 2. Materials and Methods

### 2.1. Animal Grouping

In order to study intramuscular fat infiltration, it would be best to use a sarcopenia animal such as the senescence-accelerated mouse prone-8 (SAMP8) [5,24,25] with a documented intramuscular fat infiltration problem. However, limited by the small size of skeletal muscles on the SAMP8 mice for the HR-pQCT regarding measurements with sufficient resolution, the animal model selected was an ovariectomized Sprague Dawley (SD) rat model as estrogen deficiency was known to elevate adipogenesis and fat content in skeletal muscles [26,27]. Approval was granted from the Animal Experimentation Ethics Committee (AEEC) of the Chinese University of Hong Kong (CUHK) (Ref: 15-158-MIS). Animals were fed with standard rat chow and allowed free cage movement. Bilateral ovariectomy or Sham operation was performed as previously described to 6-month-old SD rats and aged for 3 additional months according to our previous protocol [28,29]. After euthanasia with an overdose of 20% pentobarbital, muscle specimens were collected from 9-month-old ovariectomized (OVX, *n* = 4) and Sham-operated (SHAM, *n* = 4) female SD rats. Sample size was estimated based on the observed difference in intramuscular fat between normal and OVX rats of 30% and a SD of 15%, with a statistical power of 80% at an α level of 5%. The gastrocnemius in the lower leg of each rat was harvested for micro-CT and histological assessments.

### 2.2. Micro-CT Analysis of Animal Muscle

The specimens were scanned at a resolution of 8.4 μm isotropic voxel size at a beam energy of 70 kVp, current of 114 μA, and 200 ms of integration time (μCT40, Scanco Medical, Brüttisellen, Switzerland). A stack of sixty slices covering 0.480 mm were scanned at the middle of each gastrocnemius muscle. The images were analyzed using the manufacturer’s evaluation program (μCT Evaluation Program v6.0, Scanco) according to a custom analysis protocol, determining the IMAT volume based on segmentation. Analyses of muscle tissues and IMAT were performed at the region of interest (ROI) consisting of a contour covering each muscle, spanning all slices of the scan. A Gaussian filter was applied to all images to reduce noise (sigma = 2.5, support = 5) as noise is known to influence the quantification of IMAT [30]. “Intra”-muscular IMAT was segmented using a lower threshold of −600 HU (Hounsfield units), which was supported by Erlandson et al. for the segmentation of fat for the evaluation of muscle and myotendinous tissue [23], and an upper threshold of 100 HU, which produced the best correlation between MD and IMAT in our preliminary study. To avoid border artefacts, the defined ROI was peeled with 3 voxels. The ROI, MD, and percentage of IMAT volume over the total ROI volume (IMAT%) were determined.

### 2.3. Histological Analysis of Animal Muscle 

After micro-CT scanning, the muscle specimens were snap-frozen until cryo-sectioning. Muscle specimens were thawed and allowed to stabilize to the temperature of the microtome (CryoStar NX70, Thermo Fisher Scientific, Waltham, MA, USA) for 20 min. Muscle specimens were fixed with an OCT embedding compound (Tissue Tek, Sakura Finetek, CA, USA) and sectioned at 8 μm, and then thaw-mounted on silane-coated glass slides. Sections were then subject to hematoxylin and eosin (H&E) staining and oil red O (ORO) staining for the evaluation of intramuscular fat infiltration as previously described [5,31]. Briefly, for ORO staining, the slides were fixed in 10% formalin and 100% propylene glycol was added after incubation and washing. The slides were then stained with ORO solution for 10 min, counterstained with hematoxylin, and mounted with glycerin jelly mounting medium. Images were acquired using a light microscope (Leica DM5500 B, Leica Microsystems GmbH, Wetzlar, Germany) at 20× magnification. For each specimen, four images were analyzed by manually selecting the space between the cells containing connective and adipose tissue using graphical software (Adobe Photoshop CS6, Adobe Systems Inc., San José, CA, USA). Positive-stained areas were then quantified by color threshold using ImageJ software (version 1.52a, Wayne Rasband National Institutes of Health, Bethesda, Maryland, USA). The same thresholding values were used for all images in the semi-quantitative evaluation with hue: 228–255, saturation: 167–255, and brightness: 0–255. The stained area was expressed as the percentage (%) area fraction averaged from four separate images.

### 2.4. Subject Recruitment

Thirty adult Chinese female subjects from 18 to 81 years old (mean = 47.3 ± 19.2 years old) who suffered from a long bone fracture (femoral or tibial shaft) were recruited at the outpatient clinic of the Prince of Wales Hospital, Shatin, Hong Kong. Subjects were requested to take the HR-pQCT measurements, and muscle and functional assessments as described below. All assessments were performed in the Bone Quality and Health Assessment Centre, Department of Orthopaedics and Traumatology, the Chinese University of Hong Kong, which is ISO 9001-certified for daily operation. The study protocol was approved by the Joint Chinese University of Hong Kong-New Territories East Cluster Clinical Research Ethics Committee (CRE-2008.530). Written informed consents were obtained from all subjects.

### 2.5. HR-pQCT Measurement of Subjects

The nonfractured-side distal tibias of all subjects were scanned by HR-pQCT (XtremeCT version I, Scanco Medical AG, Brüttisellen, Switzerland) using the standard patient protocol with an isotropic voxel size of 82 μm as our previous protocol [20]. The subject’s leg was immobilized in a carbon fiber cast fixed within the scanner gantry. Proper positioning was ensured to minimize motion artefacts during scanning. A 2D scout view of the distal tibia was used to define the ROI. A reference line was placed at the end plate of the tibia. Measurements of 110 slices were acquired 22.5 mm proximal to the reference line. Each scan was carefully examined by the operator for motion artefacts, and up to two repeated scans at each site were performed in the case of significant motion artefacts. All images were graded for motion artefacts according to a visual grading system [32]. The short-term reproducibility of the vBMD parameter, expressed as the coefficient of variance, ranges from 0.38% to 1.03%, and the short-term reproducibility of microarchitectural parameters ranges from 0.80% to 3.73% [20,33].

### 2.6. Evaluation of Soft Tissue and Intramuscular Fat Content by HR-pQCT

The images were analyzed using the manufacturer’s evaluation program (μCT Evaluation Program v6.0, Scanco) according to the Soft Tissue Analysis (STA) protocol (version 2.0) for the determination of the muscle ROI based on an earlier version of the STA protocol (version 1.0), previously described by Erlandson et al. to evaluate lean muscle and myotendinous tissue [23]. The method, however, only detects large areas of muscle and adipose tissue, leaving the IMAT undetected. In our modified protocol, we applied the STA protocol to optimize the thresholding method to isolate the muscle as the ROI for the segmentation and separation of lean and fat tissue within the muscle group for “intra”-muscular evaluation. Briefly, images were downscaled to 164 μm to reduce noise, followed by the exclusion of bone and skin from the analysis. The muscle and adipose tissues were defined by an iterative program using carefully adjusted thresholds to plant seed volumes in the soft tissues. By region growing, the soft tissue regions were defined after 20 iterations. The algorithm generated the contours of the muscle and adipose tissues and evaluated the total muscle volume (MV) in mm^3^ and muscle density (MD) derived from the average attenuation coefficients in mgHA/mm^3^ or HU. The default thresholds used were recommended by the manufacturer, 100 to 600 HU for muscle and −600 to −200 HU for fat (Figure 1). In our modified protocol, the threshold used for the segmentation of the IMAT was manually adjusted to −20 to 100 HU, as this threshold gave the most realistic results based on visual inspection (Figure 1). The settings for the Gaussian filters (sigma = 2.5, support = 5) remained unchanged.

### 2.7. Muscle Strength and Functional Assessment

The quadriceps and hamstring muscle strength on the nonfracture side were measured by instructing the subjects to perform an active extension of the knee joint in a sitting position with both feet free from the ground, and the hip and knee joint flexed at 90°. The peak isometric forces of the knee extension were measured by a dynamometer attached at the malleoli level. Measurements were repeated thrice and the maximum force was used for analysis as our previous protocol [34]. A chair rising test (CRT) was taken by each subject on the well-reported force plates (Leonardo Mechanograph, Novotec Medical, Pforzheim, Germany). Briefly, each subject was instructed to stand up until the knees were straight and immediately sat down for five consecutive times. Measurements were recorded by the built-in software and potential associations with HR-pQCT measurements were evaluated.

### 2.8. Statistical Analysis

An independent Student’s *t*-test was conducted to analyze the data of the SHAM and OVX rats. The potential association between IMAT%, MD, and ORO area was evaluated with Pearson’s correlations. For the clinical part, linear regression analysis was performed and the coefficient of determination (R^2^) was calculated to analyze the relationship between age and MD, and between age and the amount of IMAT in the defined muscle regions. The Pearson correlation coefficient (r) was calculated to determine the degree of correlation between the IMAT, MD, and chair rise time. All statistical analyses were performed using Prism (version 6.01, GraphPad Software, Inc., San Diego, CA, USA). *p* < 0.05 was regarded as statistically significant, and the Shapiro–Wilk normality test was used where necessary.

## 3. Results

### 3.1. IMAT and MD Reflect Intramuscular Fat Content and Muscle Quality in Rats

Micro-CT results showed that IMAT% was significantly higher (*p* = 0.03) in OVX rats than in SHAM rats (Figure 2A). However, no significance was detected for MD between the two groups (Figure 2B). Histologically, from the H&E staining, the OVX group presented more intramuscular fat tissue than the SHAM group (Figure 3A). Oil red O staining was used to analyze the fat infiltration in skeletal muscle tissue of the SHAM and OVX rats. The OVX group showed a 160% higher ORO area compared to the SHAM group (*p* = 0.03) (Figure 3B,C). The IMAT% evaluated by the microCT was shown to be 39% higher in the OVX group, and was strongly associated with the ORO-positively stained area (r = 0.898, *p* = 0.002) (Figure 4).

### 3.2. HR-pQCT Parameters Associated with Intramuscular Fat Content and Muscle Performance 

Of the HR-pQCT parameters produced, a moderate positive correlation was detected between MV and MD (r = 0.642, *p* < 0.001), and between MV and IMAT volume (r = 0.618, *p* < 0.01). Moderate negative correlations were detected between MD and IMAT% (r = −0.640, *p* < 0.001).

The age of the participants was found to be negatively associated with HR-pQCT parameters, including MV (r = −0.479, *p* < 0.01), IMAT/MV (r = 0.620, *p* < 0.01), MD (r = −0.763, *p* < 0.05), and MCSA (r = −0.479, *p* < 0.05), and positively correlated with IMAT% (r = 0.559, *p* < 0.01), all with statistical significance (Table 1). Age was also found to correlate with all muscle strength and performance outcomes, with a moderate to strong correlation detected with Pmax (r = −0.608, *p* < 0.001) of the chair rise test, and both the quadriceps and hamstring strength (r = −0.686 and r = −0.638, respectively, both at *p* < 0.01, Table 2).

HR-pQCT-evaluated soft tissue parameters were found to associate moderately with functional outcomes including TV vs. P_max_ (r = 0.631, *p* < 0.01), MD vs. P_max_ (r = 0.444, *p* < 0.05), MD vs. quadriceps strength (r = 0.588, *p* < 0.01), and MD vs. hamstring strength (r = 0.565). Furthermore, the fat-related parameter of IMAT% was found to be statistically associated with rise time (r = 0.671, *p* < 0.01) and quadriceps strength (r = −0.559, *p* < 0.001), all with statistical significances (Table 2).

## 4. Discussion

The current study attempted to verify and provide scientific evidence to support the utilization of HR-pQCT for the evaluation of muscle quality and the detection of intramuscular fat infiltration, which is potentially applicable for the diagnosis or identification of sarcopenia for personalized intervention decisions. It is highly clinically significant for the prevention of falls and fragility fractures during musculoskeletal ageing. From the results of evaluating animal samples and human scans, CT parameters correlated well with the intramuscular fat content and muscle performance. Our results showed that HR-pQCT not only has the advantage of providing added structural information of bone tissue over the course of the ageing process [20], but also the capability of evaluating the quality of muscle down to the smallest changes in muscle density and predicting intramuscular fat infiltration. Therefore, it is a feasible method to be utilized clinically for bone and muscle evaluation with just a single scan.

The initial aim of the pre-clinical study was to implement the IMAT protocol using the upper threshold with the best results. However, the IMAT segmentation using this threshold (100 HU) resulted in unrealistic results, as shown in Figure 1C. This was most likely due to the lower resolution of the HR-pQCT (82 μm) compared to the μCT images of the rat leg (8.4 μm), causing high interference of partial volumes. Additionally, the human images were made with a different scanner, which could influence the linear attenuation measured. Therefore, a new upper threshold of −20 HU was determined visually, which resulted in a strong correlation with the measured MD. Nevertheless, the accuracy of this threshold remains subjective, as is shown by the discrepancy of the segmented adipose tissue area (39% higher in OVX) and the histological lipid deposition area (160% higher in the OVX), indicating an underestimated intramuscular fat content by the microCT; therefore, it should be further validated. A similar validation study performed in the pre-clinical study could also be performed on human subjects, but retrieving muscle biopsies from patients can be challenging. An alternative could be post-mortem studies, although muscle tissue quickly deteriorates after death.

Myosteatosis, or intramuscular fat infiltration, has attracted increasing attention recently due to the escalating aging population, and it is recognized as one of the many causes of sarcopenia. Many factors may cause this to happen during the aging process, including a lack of physical activities or mechanical stimulation [35], changes in the adipogenic or myogenic properties of muscle stem cells [5], and estrogen deficiency [36]. Muscle quantification and evaluation can be performed with various imaging modalities including magnetic resonance imaging (MRI), dual-energy X-ray absorptiometry (DXA), and peripheral quantitative computed tomography (pQCT), each with their pros, cons, and limitations of accessibility [37]. Until a further consensus of standardized protocol by MRI or another imaging protocol is reached for the evaluation of intermuscular or intramuscular fatty content [17], HR-pQCT could be one possible alternative that has the advantage of producing high-resolution bone, muscle, and adipose tissue data with a very low radiation dosage (~3 μSv per scan) that can be performed relatively quickly [38]. However, little research has been conducted to evaluate its use in skeletal muscle. Erlandson et al. compared the muscle-related parameters produced by the HR-pQCT against the more conventional pQCT and found that both muscle density and cross-sectional areas correlated well [23,39], and they suggested that the HR-pQCT can be used to evaluate the amount of myotendinous tissue (Mt) and quantification of muscle density (MD). Their conclusion is also supported by our current study, indicating that the HR-pQCT parameters of the muscle density (MD) and intramuscular IMAT% correlated well with age and muscle functions. Recently, Hildebrand et al. also demonstrated good precision and repeatability of the HR-pQCT against the more traditional method of pQCT that also supports its potential clinical application [40]. Furthermore, this is supported with our animal study showing that the IMAT% evaluated by the HR-pQCT had a moderately strong correlation (r^2^ = 0.806) with the quantification of intramuscular lipids by Oil Red O histology. Although HR-pQCT is disadvantageous to MRI, limited to its ability to differentiate various soft tissues [23], from our findings in this study, we recommend the use of this imaging modality as a rapid evaluation of muscle quality and objective quantification of skeletal muscle at the peripheral for sarcopenia assessment, while gathering bone micro-architectural parameters is important for osteoporosis and fracture risk prediction [41] in order to tackle two disease in one go for the prevention of fragility fracture.

The chair rise time has been shown to be a good predictive tool for sarcopenia in elderly women by Pinheiro et al. [42]. Additionally, Patel et al. [43] showed that women without sarcopenia completed the chair rise test faster than women with sarcopenia. The results from this study showed a moderate positive relationship between the chair rise time and IMAT, suggesting that the IMAT content is higher in patients with a longer chair rise time. It should be noted that the studies by Pinheiro et al. and Patel et al. only included elderly participants, while this study included young and elderly patients. Therefore, IMAT content is predictive of physical performance during the age-related changes in skeletal muscles.

The literature has shown that an increase in IMAT is not only related to sarcopenia [14] but also to an increase in BMI in both young and elderly populations [44]. Any changes in MD, IMAT, and chair rise time in this study can be related to either age, BMI, or a combination of these two factors. One limitation of our study is that anthropometric measures of fractured subjects were not collected nor taken into account. Nevertheless, as this was a study to investigate the potential of the IMAT protocol on HR-pQCT as an imaging modality, these factors did not have a major impairment to the success of this study.

## 5. Conclusions

The modified HR-pQCT evaluation protocol produced MD and IMAT results that were associated with age and physical performance measures and were of good predictive value for the progression of myosteatosis. The protocol was validated on animal skeletal muscle samples that further support the clinical application for the rapid evaluation of muscle quality and the objective quantification of skeletal muscle at the peripheral for sarcopenia assessment.

## Figures and Tables

**Figure 1 jpm-12-01016-f001:**
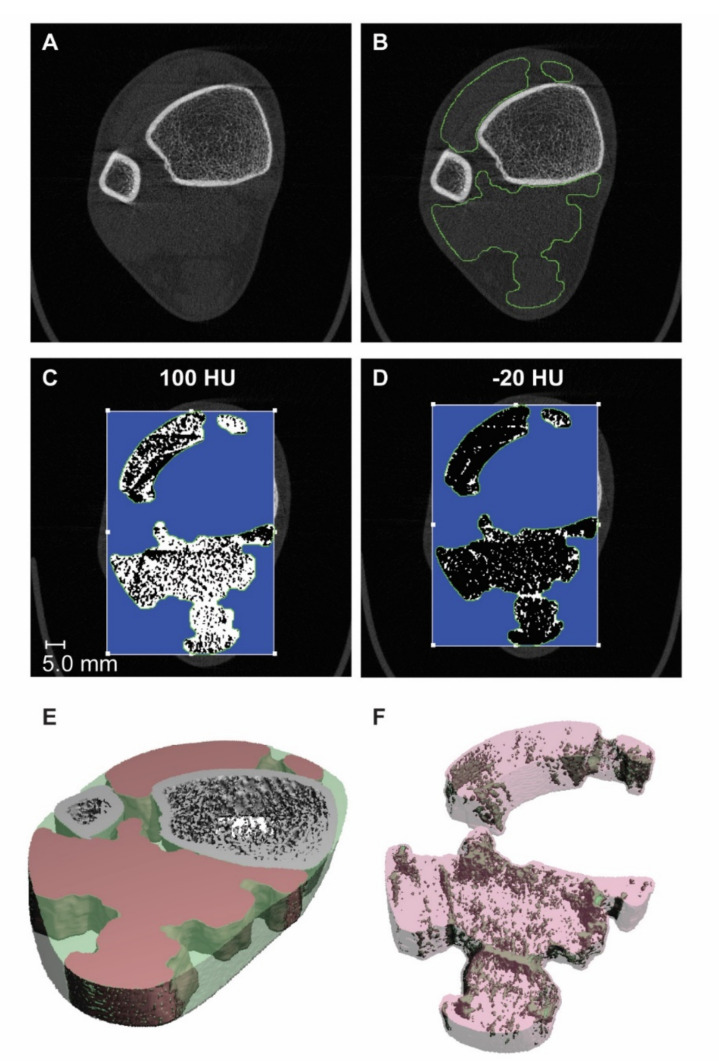
Representative images showing representation of the IMAT analysis protocol and the resulting 3D images. (**A**) In the original image, the ROI was selected using the muscle contours determined by the STA protocol, which is (**B**) depicted in green. (**C**) The intended segmentation threshold with an upper threshold of 100 HU did not produce a realistic result and was, therefore, rejected. (**D**) The upper threshold of −20 HU was defined manually. (**E**) The STA protocol produces a 3D segmentation, with the bones depicted in grey, the muscles in red, and fat in transparent green. (**F**) Application of the IMAT segmentation produces a 3D image, as shown with the IMAT depicted in green and the muscle depicted in transparent red.

**Figure 2 jpm-12-01016-f002:**
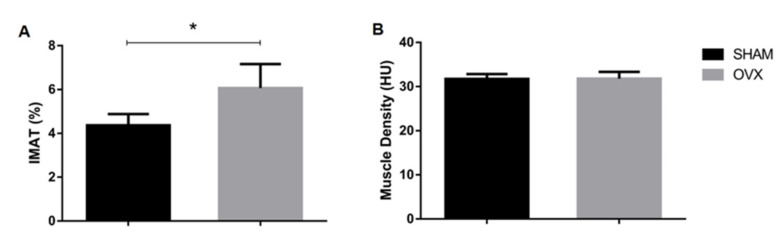
(**A**) IMAT% was significantly higher in OVX rats than in SHAM rats (* *p* < 0.05). (**B**) No significance was detected for MD between the two groups.

**Figure 3 jpm-12-01016-f003:**
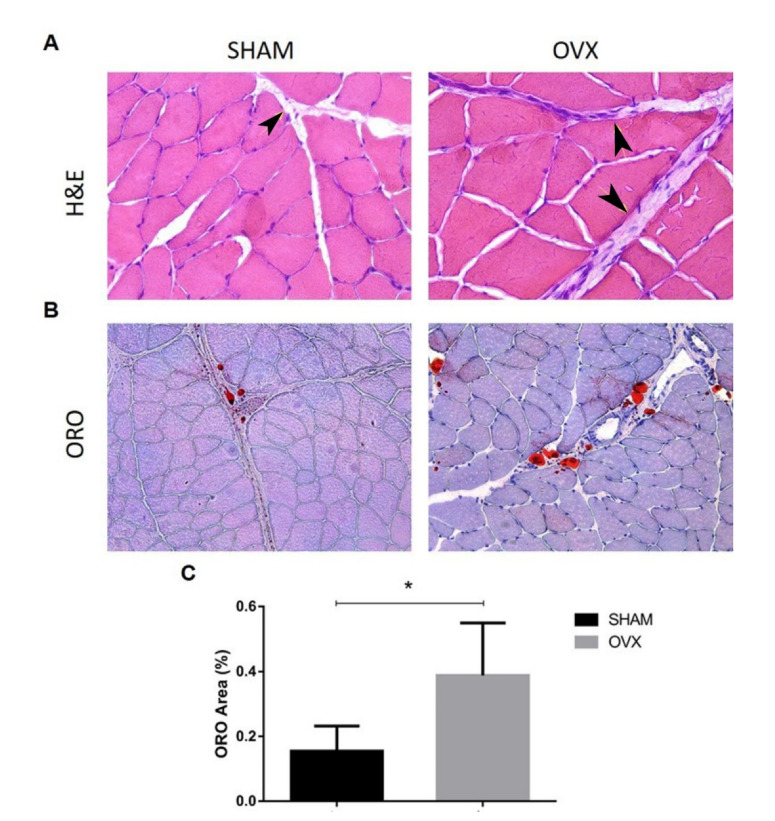
Morphological differences and fat infiltration of skeletal muscles in SHAM and OVX rats. (**A**) The OVX group presented more intramuscular fat tissue than the SHAM group by H&E taken at 20× magnification, where fat tissues are indicated by black arrows. (**B**) The OVX group showed a higher ORO signal (red area) than the SHAM group. (**C**) Quantitative analysis revealed that the ORO area of the OVX group was significantly higher (* *p* < 0.05, Student’s T-test).

**Figure 4 jpm-12-01016-f004:**
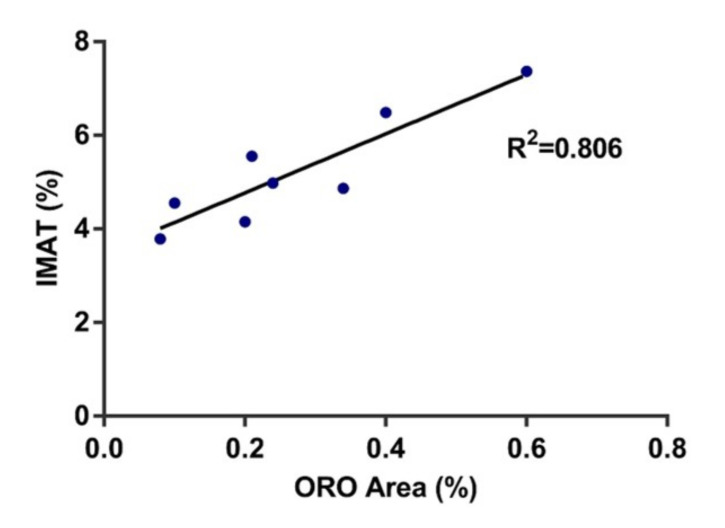
MicroCT fat-related parameter of IMAT% was highly correlated with ORO-positively stained area (r = 0.898, r^2^ = 0.806; *p* < 0.05).

**Table 1 jpm-12-01016-t001:** Correlations between age and various muscle-related parameters produced by HR-pQCT.

	Tibia CSA [cm^2^]	Total.Volume	Muscle.Volume	MV/TV	IMAT.Volume	IMAT/MV	Muscle.Density	Fat.Density	MCSA	IMAT.V	MUS.V	IMAT%
Age	−0.504	−0.380	−0.479 *	−0.208	0.309	0.620 **	−0.763 **	−0.074	−0.479 *	0.070	−0.483 *	0.559 **

Significant correlations were found in MV, IMAT/MV, MD, and IMAT%. * and ** represent significant correlations at the 0.05 and 0.01 levels (2-tailed), respectively.

**Table 2 jpm-12-01016-t002:** Correlations between various muscle-related parameters produced by HR-pQCT and functional parameters of the subjects.

	P_max_	P_max_ (Fracture)	P_max_ ((Normal)	Total Time	Time per Test	Rise Time	Quadriceps Strength	Hamstring Strength
Age	−0.608 **	−0.406	−0.633 **	0.507 *	0.514 *	0.434 *	−0.686 **	−0.638 **
Tibia CSA	0.615 *	0.313	0.653 *	−0.112	−0.304	−0.099	0.468	0.454
TV	0.631 **	0.497 *	0.619 **	−0.104	−0.306	−0.085	0.438 *	0.330
MV	0.317	0.207	0.338	−0.413 *	−0.354	−0.476 *	0.245	0.282
MV/TV	−0.236	−0.203	−0.219	−0.377	−0.097	−0.441 *	−0.078	0.064
IMAT.V.	−0.070	0.002	−0.095	0.211	0.112	0.168	−0.385	−0.279
IMAT/MV	−0.214	−0.089	−0.249	0.452 *	0.298	0.525 *	−0.560 **	−0.429 *
MD	0.444 *	0.262	0.489 *	−0.704 **	−0.625 **	−0.610 **	0.588 **	0.565 **
FD	−0.306	−0.296	−0.267	−0.192	0.152	−0.156	−0.213	−0.024
MCSA	0.317	0.207	0.338	−0.413 *	−0.354	−0.476 *	0.245	0.282
● IMAT.V	0.003	−0.067	0.052	0.055	0.024	0.086	−0.273	−0.080
● MUS.V	0.301	0.190	0.327	−0.486 *	−0.384	−0.515 *	0.254	0.291
● IMAT%	−0.245	−0.252	−0.200	0.436 *	0.311	0.671 **	−0.559 **	−0.380

Muscle Density (MD) and IMAT% were found to be the most predictive parameters with statistically significant correlations. * and ** represent significant correlations at the 0.05 and 0.01 levels (2-tailed), respectively. Abbreviations: CSA = cross-sectional area, TV = tissue volume, MV = muscle volume, D = density, AT = adipose tissue. ● Designates measurements from the modified protocol.

## Data Availability

The data presented in this study are available on request from the corresponding author. The data are not publicly available, due to restrictions from institutional ethics approval.

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
