# Peer review of "HR-pQCT for the Evaluation of Muscle Quality and Intramuscular Fat Infiltration in Ageing Skeletal Muscle"

_jpm, 2022, doi:10.3390/jpm12061016_

Round 1

Reviewer 1 Report

Myosteatosis and sarcopenia developing as a result is a very important condition for the elderly population and affects the quality of life. Therefore, its detection and treatment is very important. Therefore, I think that this study will contribute to the literature.

I think that comparing the mentioned technique/method with the existing techniques in terms of clinical applicability and cost-effectiveness will have a positive effect on the article.

I think that this well designed and written article will guide other experimental studies and clinical studies.

However, I would like to state that the discussion section of the study should be improved. By showing more reference studies, the clinical applicability of this method can be commented on.

Author Response

Reviewer 1

I think that comparing the mentioned technique/method with the existing techniques in terms of clinical applicability and cost-effectiveness will have a positive effect on the article.

RESPONSE: Thank you for your comment.  We agree that cost effectiveness of using such method for the prevention of sarcopenia or subsequent fractures related to sarcopenia would be very valuable.  However, it would involve a much more stringent study design involving public health and the involvement of public health authorities in order or us to make a meaning conclusion. This part is well beyond the reach of our research group at this stage.

I think that this well designed and written article will guide other experimental studies and clinical studies.

RESPONSE: However, I would like to state that the discussion section of the study should be improved. By showing more reference studies, the clinical applicability of this method can be commented on.

Thank you for the comment.  As the detection of intramuscular fat using HR-pQCT is very novel and not many studies are available, the discussion section has been substantiated with one relevant recently published article at line 340 to 342.  Clinical applicability can only be commented upon our own user experience targeting to tackle 2 diseases at one go.  Revision is made at line 349 to 350.

Reviewer 2 Report

  1. This manuscript represents good research finding and has adequate scientific merit. The argument in the introduction and discussion complies with the manuscript objectives. They manage to develop a modified HR-pQCT evaluation protocol for the detection and analysis of IMAT, and the protocol was validated in animal and clinical studies.

    However, minor amendments are suggested to be done in the methodology and results sections. My comments are stated below:

    a) In my opinion, the authors should briefly show the sample size calculation for both animal (n = 4 per group) and clinical studies (total = 30 subjects).

    b) The authors should give justification of choosing subjects that are suffered from femoral or tibial shaft fracture?. Is there any relation with gastrocnemius muscle?. Is the big age range of the subjects between 18 to 81 years old is acceptable to reflect aging skeletal muscle?.

    c) For statistical analysis, the name of the normality test used should be included by the authors.

    d) For Figure 3 A and B, it is highly advisable to include the scale information and state the image magnification. The yellow arrows are not clearly seen. 

    e) All figures and tables should be clearly explainable without referring to the text. 

Author Response

This manuscript represents good research finding and has adequate scientific merit. The argument in the introduction and discussion complies with the manuscript objectives. They manage to develop a modified HR-pQCT evaluation protocol for the detection and analysis of IMAT, and the protocol was validated in animal and clinical studies.

However, minor amendments are suggested to be done in the methodology and results sections. My comments are stated below:

  1. a) In my opinion, the authors should briefly show the sample size calculation for both animal (n = 4 per group) and clinical studies (total = 30 subjects).

Sample size estimation is added to line 115 to 117.  For the association study in human subjects using regression analysis, sample size is larger the better and thus estimation was not relevant and therefore not performed.

  1. b) The authors should give justification of choosing subjects that are suffered from femoral or tibial shaft fracture?. Is there any relation with gastrocnemius muscle?. Is the big age range of the subjects between 18 to 81 years old is acceptable to reflect aging skeletal muscle?.

Thank you for your comment, fracture cases included all causes including traumatic accidents.  The study was intended to investigate the scanning and evaluation protocol in relations to measured physical performance.  Therefore, investigating the muscle quality of the gastrocnemius and cause of fracture was beyond the scope of the study.  As shown by our results, correlation of muscle quality and physical performance does correlate nicely and thus we believe the age range is suitable for such conclusion being made.  These are described in line 355 to 361, and limitation is added to state fracture patients were studied (line 366).

  1. c) For statistical analysis, the name of the normality test used should be included by the authors.

The missing information is added to line 233 to 234.

  1. d) For Figure 3 A and B, it is highly advisable to include the scale information and state the image magnification. The yellow arrows are not clearly seen.

Relevant changes have been made on Figure 3 and line 250 to 253.

  1. e) All figures and tables should be clearly explainable without referring to the text.

Thank you for pointing out the flaw, figure captions are updated to enhance readability.